# BiXT: Perceiving Longer Sequences With Bi-Directional Cross-Attention Transformers

## Abstract

We present a novel bi-directional Transformer architecture (BiXT) for which computational cost and memory consumption scale linearly with input size, but without suffering the drop in performance or limitation to only one input modality seen with other efficient Transformer-based approaches. BiXT is inspired by the Perceiver architectures but replaces iterative attention with an efficient bi-directional cross-attention module in which input tokens and latent variables attend to each other simultaneously, leveraging a naturally emerging attention-symmetry between the two. This approach unlocks a key bottleneck experienced by Perceiver-like architectures and enables the processing and interpretation of both semantics ('what') and location ('where') to develop alongside each other over multiple layers – allowing its direct application to dense and instance-based tasks alike. By combining efficiency with the generality and performance of a full Transformer architecture, BiXT can processes longer sequences like point clouds or images at higher feature resolutions. Our tiny model variant achieves accuracies up to $82.0\%$ for classification on ImageNet1K with no modality-specific internal components, and performs competitively on semantic image segmentation (ADE20K) and point cloud part segmentation (ShapeNetPart) even against modality-specific methods.

## 1 Introduction

Much of the data we obtain when perceiving our environment can be interpreted via a division into '*what*' and '*where*'. If we consider for example the image pictured in Figure 1 on the left, we can easily describe its content via 'what' we see – the building, sky and a flag. If we were to draw conclusions on a more fine-grained level though, we would likely include more specific descriptions like "lower left corner" referring to their positions within the image – the 'where'. In other words, 'where' denotes the actual geometric location of the individual elements (e.g. pixels) and 'what' the semantic entities (e.g. objects) that collectively describe the data as a whole. Note that this similarly applies to many other modalities, like point clouds or even language where we form words via letters that together have a certain meaning.

Thanks to the few structural constraints placed on the input data paired with high performance, Transformers (Vaswani et al., 2017) have shown great capabilities in extracting both 'what' and 'where' for a range of input modalities, giving rise to significant advances across various fields such as Natural Language Processing (Devlin et al., 2019) and Computer Vision (Dosovitskiy et al., 2021; Touvron et al., 2021; 2022). However, their success comes at the high cost of scaling quadratically in memory and time with the input length, practically excluding their use on larger input data like point clouds or high-resolution images when resources are limited.

Several approaches have since been proposed to increase their efficiency, either by changing how the computationally expensive self-attention operation is realized (Wang et al., 2020; Shen et al., 2021) or by exploiting the domain-specific structure of their data input (Parmar et al., 2018; Ho et al., 2019; Qiu et al., 2020; Tu et al., 2022). However, these approaches have a tradeoff of reducing the Transformer's performance or limiting its application to only one specific type of input (El-Nouby et al., 2021).

In an attempt to preserve the generality by not imposing additional constraints on the input data, Jaegle et al. (2021) employ a small set of latent vectors as a bottleneck to extract the 'what' via one-sided (iterative) cross-attention – and require an additional decoder to draw conclusions about

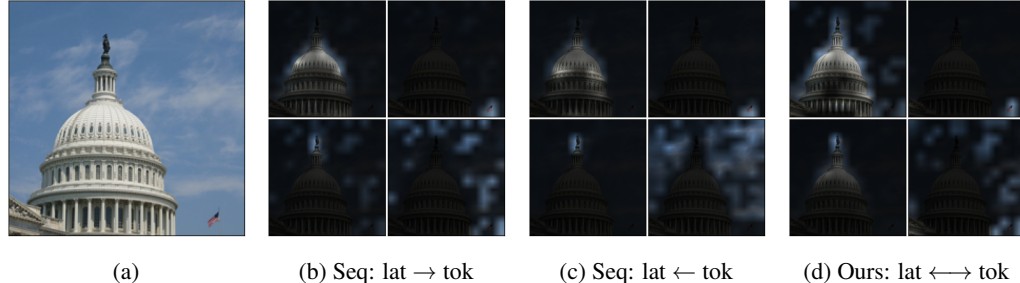

|  |  |  |  |
|---|---|---|---|
| (a) | (b) Seq: lat $\rightarrow$ tok | (c) Seq: lat $\leftarrow$ tok | (d) Ours: lat $\longleftrightarrow$ tok |

Figure 1: **Emerging patterns when attending both ways.** (a) Input image. (b) depicts the areas of the image that 4 different latents attend to, while (c) inversely shows which image regions attend to these latents (transformed into the same coordinate system for ease of interpretation). (d) displays which areas & latents are symmetrically attended to using our proposed bi-directional cross-attention.

'where' (Jaegle et al., 2022). While achieving linear complexity w.r.t. the input length, these 'Perceiver' architectures require between 360 - 707 GFLOPs to achieve around $78\%$ accuracy on ImageNet1K – results that recent ViT variants (Touvron et al., 2021; 2022) are able to obtain at a fraction of the compute. One possible explanation for this discrepancy is that the effective working memory of Perceiver architectures is strictly limited to the latents which therefore need to compensate via increased computation, whereas conventional Transformers like ViTs leverage the (larger) number of tokens across several layers. This raises an important question: Are the appealing individual properties of these two methods mutually exclusive, or can we in fact have *the best of both worlds?*

In this paper, we set out to affirm the latter. We demonstrate that a small set of latent vectors appropriately combined with layerwise simultaneous refinement of both input tokens and latents makes it possible to pair the high performance and architectural simplicity of ViTs with the linear scaling of Perceivers – outperforming both ViT and Perceiver in settings where compute is limited.

We start off by investigating a naïve approach: sequentially applying cross-attention to refine 'what' and 'where', one after the other. We discover that approximately symmetric attention patterns naturally emerge between latents and tokens even when both are provided with complete flexibility. In other words, for most latents ('what') that pay attention to particular tokens ('where'), these tokens in turn pay attention to exactly these latents (see Figure 1 and Section 3.1). Not only does this intuitively make sense – objects need to know 'where' they are located in the image, and image locations need to know 'what' objects are located there – it more importantly offers us a unique opportunity to save FLOPs, memory and parameters.

As we will demonstrate in Section 2, this symmetry means we only need to compute the attention matrix once, reducing the required parameters by $\sim 1/3$, to facilitate a symmetric, bi-directional information exchange via our proposed *bi-directional cross-attention*. Integrated into our bi-directional cross-attention Transformer architecture (BiXT), this forms a flexible and high-performing yet efficient way to process different input modalities like images and point clouds on a variety of instance-based (e.g. classification) or dense tasks (e.g. point cloud part segmentation) – all while scaling linearly w.r.t. the input length.

In summary, our main contributions include the following:

1. We introduce a novel bi-directional cross-attention Transformer architecture (*BiXT*) that scales linearly with the input size in terms of computational cost and memory consumption, allowing to process longer sequences like point clouds or images at higher resolution.
2. We propose *bi-directional cross-attention* as an efficient way to establish symmetric information exchange that requires computation of the attention matrix only *once* and reduces the involved parameters by $\sim 1/3$, leveraging a naturally emerging symmetry in cross-attention and showing significant improvements over uni-directional iterative methods.
3. We analyse BiXT's advantage of processing longer sequences across a number of tasks using different input modalities and output structures in settings with limited computational resources – achieving e.g. accuracies up to $82\%$ for classification on ImageNet1K with tiny models and no modality-specific internal components, and performing competitively for semantic image segmentation on ADE20K and point cloud part segmentation on ShapeNetPart even among modality-specific approaches.

4. We further provide insights into BiXT's extendibility: Thanks to its simple and flexible design, modality-specific components can easily be incorporated in a plug-and-play fashion should the need arise – further improving results while trading off generality.

## 2 PERCEPTION VIA BI-DIRECTIONAL CROSS-ATTENTION

We start this section by briefly revisiting the concept of attention before moving on to presenting our proposed *bi-directional cross-attention* methodology, followed by its use within our BiXT architecture (Figure 2). Please note that we define the concepts using single-head attention for brevity instead of the actually employed multi-head attention (MHA), and all methods directly generalize to MHA.

### 2.1 BACKGROUND: THE ATTENTION MECHANISM AND ITS COMPLEXITY

While self-attention has recently gained great popularity through its use in the Transformer architecture (Vaswani et al., 2017), we will start from a slightly more general point of view: Given a source sequence $\mathcal{S} \in \mathbb{R}^{N \times D_{\mathcal{S}}}$ and a target sequence $\mathcal{T} \in \mathbb{R}^{M \times D_{\mathcal{T}}}$, attention aims to refine $\mathcal{T}$ by exhaustively discovering pairwise correlations between all elements of both sequences and integrating information from the source components of interest into the target.

Formally, $\mathcal{S}$ is linearly projected into two $D$-dimensional representations using learnable matrices – yielding a *key* $\boldsymbol{K}_{\mathcal{S}} \in \mathbb{R}^{N \times D}$ and *value* $\boldsymbol{V}_{\mathcal{S}} \in \mathbb{R}^{N \times D}$ – while $\mathcal{T}$ is projected into one $D$-dimensional representation to obtain the *query* $\boldsymbol{Q}_{\mathcal{T}} \in \mathbb{R}^{M \times D}$. These representations are then used to compute the attention-based target refinement as

$$\Delta_{\mathcal{T}}^{\text{attn}} = \text{attn}\left(\boldsymbol{Q}_{\mathcal{T}}, \boldsymbol{K}_{\mathcal{S}}, \boldsymbol{V}_{\mathcal{S}}\right) = \text{softmax}\left(\frac{\boldsymbol{Q}_{\mathcal{T}} \boldsymbol{K}_{\mathcal{S}}^{\mathsf{T}}}{\sqrt{D}}\right) \cdot \boldsymbol{V}_{\mathcal{S}}, \tag{1}$$

with the scaled dot product $\bar{\boldsymbol{A}}_{\mathcal{T},\mathcal{S}} = {}^{1}\!/\!\sqrt{D}\left(\boldsymbol{Q}_{\mathcal{T}} \boldsymbol{K}_{\mathcal{S}}^{\mathsf{T}}\right) \in \mathbb{R}^{M \times N}$ representing the scaled pairwise similarity between target and source elements. This concept is commonly referred to as *cross-attention* between target $\mathcal{T}$ and source $\mathcal{S}$. If a representation itself is to be refined given the context within, i.e. source and target are identical ($\mathcal{S} = \mathcal{T}$), Equation (1) reduces to the well-known *self-attention* where the triplet key, query and value are all generated as a function of the same sequence.

Note that computing the similarity matrix $\bar{\boldsymbol{A}}_{\mathcal{T},\mathcal{S}}$ has computational complexity $\mathcal{O}(NM)$. For self-attention used in Transformers where $\mathcal{T} = \mathcal{S}$ and hence $M = N$, this yields quadratic complexity $\mathcal{O}(N^2)$ w.r.t. the input sequence length $N$, prohibiting its use on longer sequences when computational resources are limited. On the other hand, if cross-attention is employed with a fixed sequence length $M = \text{const}$, the complexity becomes linear $\mathcal{O}(N)$.

### 2.2 BI-DIRECTIONAL CROSS-ATTENTION

Reducing the complexity of attention from quadratic to linear without impairing performance or adding constraints w.r.t. input modalities is one of the main aspects of this work. We build our approach on the previously introduced notion that most perceptual data can be interpreted as 'what' and 'where' – and both need to pay attention to the other for optimal information exchange. We represent the 'what' via a small set of $M$ learnable *latent vectors* and the 'where' via an input-dependent sequence of $N$ *tokens*, respectively denoted via the subscripts $_{\text{lat}}$ and $_{\text{tok}}$ in the following and with $M \ll N$. Naïvely, one could simply apply two individual cross-attention operations sequentially – first querying information from one side and then the other by creating two *query-key-value* triplets. However, our analyses presented in Section 3.1 show that symmetric tendencies in the attention patterns between latents and tokens naturally emerge during training, offering a chance to further reduce the computational requirements and to increase efficiency via our *bi-directional cross-attention* as follows.

We start by creating *reference-value* pairs $\boldsymbol{R}_{\text{lat}} \in \mathbb{R}^{M \times D}, \boldsymbol{V}_{\text{lat}} \in \mathbb{R}^{M \times D}$ and $\boldsymbol{R}_{\text{tok}} \in \mathbb{R}^{N \times D}, \boldsymbol{V}_{\text{tok}} \in \mathbb{R}^{N \times D}$ via learnable linear projection from the latent vectors and tokens, respectively. Leveraging symmetry to create bi-directional information exchange, pairwise similarities between latents and tokens are then computed via a scaled dot product as

$$\bar{\boldsymbol{A}}_{\text{lat,tok}} = \left(\frac{\boldsymbol{R}_{\text{lat}} \boldsymbol{R}_{\text{tok}}^{\mathsf{T}}}{\sqrt{D}}\right) = \bar{\boldsymbol{A}}_{\text{tok,lat}}^{\mathsf{T}}, \tag{2}$$

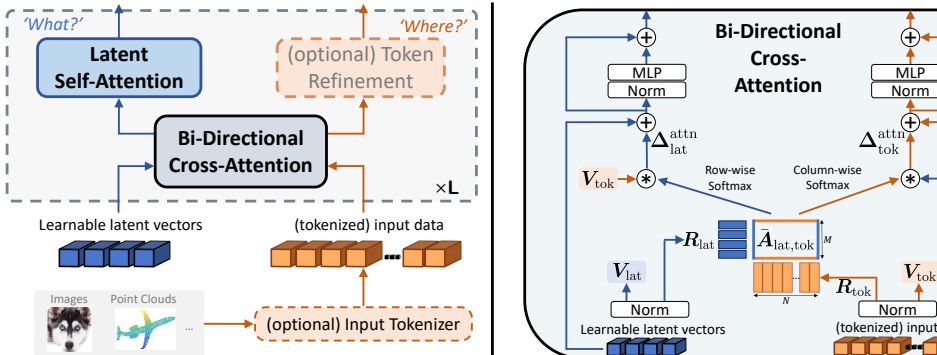

Figure 2: **BiXT architecture.** (left) Input data passing through one layer of our Bi-Directional Cross-Attention Transformer. (right) Internal structure of proposed efficient bi-directional cross-attention.

which is in turn used to obtain the attention-based refinement for the both latents and tokens via

$$\Delta_{\text{lat}}^{\text{attn}} = \text{softmax}\big(\bar{\boldsymbol{A}}_{\text{lat,tok}}\big) \cdot \boldsymbol{V}_{\text{tok}} \qquad \text{and} \qquad \Delta_{\text{tok}}^{\text{attn}} = \text{softmax}\big(\bar{\boldsymbol{A}}_{\text{tok,lat}}\big) \cdot \boldsymbol{V}_{\text{lat}}. \tag{3}$$

Note that in addition to providing linear scaling w.r.t. to the input length $N$, Equation (2) requires evaluating the most computationally-expensive operation, namely the similarity matrix ($\mathcal{O}(MN)$), only **once** and allows simultaneous refinement of latents and tokens as defined in Equation (3). The implicit reuse of the *references* as both *query* and *key* further reduces the parameter count of the linear projection matrices by $1/3$ compared to naïve sequential cross-attention.

## 2.3 BiXT – Bi-directional cross-attention transformers

Figure 2 (left) illustrates the individual components that make up our BiXT architecture. BiXT is designed in a simple symmetric, ladder-like structure allowing 'what' (latent vectors) and 'where' (tokens) to simultaneously attend to and develop alongside each other – making it equally-well suited for instance-based tasks like classification and dense tasks like semantic segmentation on a variety of input modalities. We start this section with a brief overview, followed by more detailed descriptions of the individual components.

**General overview.** The raw input data is first passed through a tokenization module which projects the data into an embedding sequence of length $N$ and optionally adds positional encodings, depending on the input modality and data structure. These tokens together with a fixed set of $M$ learnable latent vectors are then passed to the first layer's bi-directional cross-attention module for efficient refinement (details depicted in Figure 2 (right) and explained below). The latents are then further refined via latent self-attention, while the tokens are either directly passed on to the next layer (default) or optionally refined by a token refinement module which could include modality-specific components. The simultaneous ladder-like refinement of 'what' and 'where' is repeated for $L$ layers, before the result is passed to task-specific output head(s). For instance-based tasks like classification, we simply average the set of latent vectors and attach a classification head to the output, while for tasks like segmentation that require outputs resembling the input data structure, the refined tokens are used.

**Efficient bi-directional information exchange.** We use bi-directional cross-attention introduced in Section 2.2 to enable $M$ latents and $N$ tokens to simultaneously attend to each other in a time and memory efficient way, provided $M \ll N$. The detailed internal structure of our module is depicted in Figure 2(right) and defined via Equations (2) and (3). Apart from the efficient bi-directional attention computation, it follows the common Transformer-style multi-head attention in terms of normalization, activations and processing via feed-forward networks (FFN) introduced by Vaswani et al. (2017) and can thus be easily implemented in modern deep learning frameworks.

Three aspects are particularly worth noting here: 1) While bi-directional attention imposes a 'hard' structural constraint of symmetry on the information exchange between tokens and latents per layer, i.e. latents and tokens symmetrically attend to each other, the actual information that is transferred is created via individual value projection matrices and thus offers flexibility in terms of content. 2) While tokens cannot directly communicate with each other as is possible when using

computationally expensive self-attention, this communication can still take place over two layers in our structure by using a latent vector as temporary storage in a token-latent-token sequence. Since the total number of latents is usually larger than the semantic concepts required to describe one data sample, we can expect this to be possible without impairing performance. 3) Compared to the naïve approach of sequential cross-attention, we empirically find that our bi-directionally-constrained architectures are more stable in training and more robust w.r.t. hyperparameter choice (also see Section 3.1).

**Latent vector refinement.**    After gathering information from the tokens, we use one multi-head self-attention operation (Vaswani et al., 2017) to further refine the information stored in the latents and provide direct information exchange with a global receptive field across latents. Note that since the number of latents $M$ is significantly smaller than the input sequence and fixed, this operation is input-length independent and not particularly resource intensive. This step is similar to Perceiver (Jaegle et al., 2021; 2022), but we only use one instead of several self-attention operations at each layer.

**Optional token refinement.**    In the majority of experiments presented in this paper, we simply pass the tokens returned by the bi-directional cross-attention to the next layer. However, our architectural structure also allows to easily include additional (e.g. data-specific) modules for further refinement in a plug-n-play manner. We will demonstrate examples of this in Section 3, where we add a local refinement component exploiting grid-shaped data for image classification (El-Nouby et al., 2021) and a data-specific hierarchical grouping module for point cloud shape classification (Ma et al., 2022).

**Positional encodings.**    We use additive sinusoidal positional encodings (Vaswani et al., 2017) to represent the structure of input data, which is more efficient than learnt encodings for variable input size. For simplicity, we follow previous works like El-Nouby et al. (2021) and create the encodings in 32 dimensions per input axis followed by a linear projection into the model's token dimension $D$. Note that this method is applicable independent of the raw data's dimensions and thus easily handles data ranging from 2D images to 3D or 6D point clouds.

**Input tokenization.**    Tokenization can be performed in various ways and is the only input modality-specific component in our architecture, akin to Perceiver-IO's input adapters (Jaegle et al., 2022). For classification experiments on image datasets like ImageNet1K (Russakovsky et al., 2015), we follow common practice and use simple linear projection to embed image patches. For point cloud data, we simply encode the 3D or 6D points directly into embedding space using our positional encoder. Note that we do not see any reason why BiXT should not be applicable to data beyond the scope of our paper, and any data processing module that outputs a set or sequence of embeddings could be used.

## 3 EXPERIMENTAL EVALUATION

The purpose of our investigations presented in the following is twofold: 1) To provide qualitative and quantitative insights into our proposed *bi-directional cross-attention* and the underlying intuition of symmetry, and 2) to demonstrate how BiXT's ability to efficiently and effectively process longer sequences positively affects various tasks. We focus the majority of our experiments around efficient architectures in the low FLOP and parameter regime, and unless otherwise stated, we use BiXT with 64 latent vectors, embedding dimension 192 and 3 heads for all attention modules – *aka* 'BiXT-tiny'.

Note that where indicative results are presented, every architecture was only run once, whereas distributional results present mean and (unbiased) std-dev of 3 randomly seeded training runs.

### 3.1 SYMMETRIC TENDENCIES EMERGE WHEN ATTENDING BOTH WAYS

We start by investigating the intuition underlying our work: When describing data like an image by asking '*what*' is in it and '*where*' things are, it intuitively makes sense that these two components are tightly interconnected, and that they will inform *aka* pay attention to each other. To this end, we set up a naïve architecture where latent vectors first query the tokens via cross-attention (CA), followed by the tokens querying the latents (i.e. using independent query-key-value triplets), before a further refinement step of the latent information via one self-attention operation – repeated over multiple layers and trained on ImageNet1K (Russakovsky et al., 2015). When looking at the resulting attention patterns depicted in Figure 1, we discover that most latents pay attention to parts of the image representing one specific 'entity' like a building ((b), top-left), a flag ((b), top-right) or parts

Table 1: **Bi-directional vs. iterative attention.** (a) Classification accuracy on ImageNet1K. All architectures use 64 latent vectors and have been trained for 120 epochs with hyperparameters individually optimized. Architectural configurations noted in brackets. †indicates sharing of all, ‡of all but the 1st layer's cross-attention parameters. Results reported as mean and (unbiased) standard deviation over 3 randomly seeded training runs (see appendix for complete results & details). (b) Point cloud shape classification on ModelNet40. BiXT without (*naïve*) and with modality-specific components in comparison to other works (Qi et al. (2017a;b); Jaegle et al. (2021); Ma et al. (2022)).

(a) ImageNet1K @ 120epochs.

| Attention | Top-1 Acc. | FLOPs | Mem. | #Param |
|---|---|---|---|---|
| *Iterative attention (Perceiver-like)* | | | | |
| Iter.‡ (sa5-d8) | $58.26 \pm 2.34$ | 1.58G | 7.17M | 19.05M |
| Iter.‡ (sa6-d7) | $54.94 \pm 5.96$ | 1.59G | 7.23M | 19.94M |
| Iter.† (sa6-d8) | $60.61 \pm 1.11$ | 1.82G | 8.25M | 22.16M |
| Iter.† (sa4-d12) | $56.03 \pm 1.02$ | 1.99G | 9.10M | 22.16M |
| Iter.† (sa1-d24) | $55.92 \pm 0.67$ | 1.79G | 8.39M | 11.93M |
| *Cross-attention variants* | | | | |
| Seq. (2-way, d11) | $71.64 \pm 0.45$ | 1.66G | 7.52M | 14.60M |
| **Bi-Dir.** (d12) | $\mathbf{72.48} \pm 0.31$ | 1.68G | 7.23M | 15.11M |

(b) ModelNet40.

| Method | OA | mAcc |
|---|---|---|
| *Naïve, point-based* | | |
| PointNet | 89.2 | 86.0 |
| Perceiver | 85.7 | – |
| BiXT (naïve) | 89.6 | 86.4 |
| *Hierarchical, point grouping, etc.* | | |
| PointNet++ | 90.7 | – |
| PointMLP | 94.1 | 91.3 |
| BiXT (+ group) | 92.5 | 89.7 |
| BiXT (+ group & hier.) | 93.1 | 90.6 |

of the sky ((b), lower-right) – supporting the notion that latent vectors represent 'things'. More interestingly however, we discover in (c) that most of these image regions (tokens) are in turn also paying attention to exactly these latent vectors – showing a roughly symmetric information exchange and providing a qualitative indication that our idea of enforcing symmetry via our bi-directional architecture might be well justified. We additionally visualize the attention patterns after replacing the naïve sequential CA through our efficient bi-directional cross-attention in (d), and the results look surprisingly similar – clearly indicating that our symmetrically constrained approach can achieve similar information exchange while being significantly more efficient.

## 3.2 ATTENTION – ITERATIVE, SEQUENTIAL OR BI-DIRECTIONAL?

We aim to provide conclusive insights about the two major advantages of our proposed bi-directional attention compared to Perceiver's iterative attention: 1) Higher performance for comparable numbers of FLOPs, and 2) Ability to optionally extend the architecture via modality-specific components. We therefore choose two tasks to do so that have also been investigated in the Perceiver paper: Image classification on ImageNet1K (Russakovsky et al., 2015) and point cloud shape classification on ModelNet40 (Wu et al., 2015).

**ImageNet classification.** To provide a fair basis for comparison, we create a range of architectural configurations with iterative attention based on the insights reported by Jaegle et al. (2021). Targeting a comparable FLOP count as our BiXT tiny, we experiment with different numbers of layers, varying numbers of self-attention operations per block and with sharing all CA parameters as well as all but the first layer's (for details, see Perceiver paper and our appendix) – yielding a total of 10 architectures based on Perceiver's iterative attention. Having optimized the hyperparameters (learning rate and schedule) for each individually, we run 3 randomly seeded training runs for the best 5 configurations and report their results after training for 120 epochs in Table 1 (a). BiXT's bidirectional cross-attention outperforms all iterative variants by a significant margin at comparable FLOP counts and proves more robust during training as indicated by the smaller std-dev, and also beats the naïve sequential CA variant. Interestingly, we find the configuration with 8 blocks and 6 self-attention layers per block (sa6-d8) to achieve best performance among the iterative variants, which aligns exactly with the 'best' configuration reported by Jaegle et al. (2021) (albeit at a much smaller scale).

**Point cloud shape classification.** To gain further quantitative insights how bi-directional attention affects processing of other modalities, we evaluate our approach on the ModelNet40 dataset (Wu et al., 2015). BiXT again clearly outperforms Perceiver in terms of overall accuracy (OA) and is even competitive to other point-based methods like the seminal PointNet (Qi et al., 2017a) (Figure 2 (b)). In contrast to iterative attention that gathers information exclusively in the latents, BiXT's simultaneous refinement of latents and tokens allows us to easily integrate data-specific modules for token refinement. To gauge the effect, we add the 'affine grouping' module from PointMLP (Ma

et al., 2022) without and with creating a hierarchical structure (i.e. point reduction). While BiXT is still outperformed by the point cloud specific PointMLP method, these optional modules help to significantly boost the accuracy by up to $3.9\%$ while trading off generality.

### 3.3 IMAGE CLASSIFICATION

**Comparison to SOTA.** Note that we focus here on efficient models in the low FLOP and/or parameter regime, with results reported in Table 2. BiXT performs favourably in its default configuration against the other 'vanilla' Transformers, outperforming both versions of DeiT by a significant margin while being significantly more efficient than Perceiver (IO). BiXT further shows strong performance when including the 'local patch interaction' module from XCiT as modality-specific token refinement – outperforming the original work as well as most other methods including more architecturally complex pyramidal architectures.

**Increasing feature resolution and input size.** We keep the patch size fixed to $16^2$ while reducing the stride of our linear patch projector to increase feature resolution (see appendix for ablation on patch sizes vs. stride). Note that our BiXT/4 model can process 3364 tokens per $224^2$ image thanks linear scaling, boosting the top-1 accuracy to $81.2\%$. Linear scaling also lets us process larger input images more efficiently – which we investigate by fine-tuning the $224^2$-trained models on $384^2$ for 30 epochs to reduce required computational resources. Increasing the input size allows us to further notably improve the accuracy by around $1.0-2.4\%$ across architectures, however at the expense of higher FLOP counts. Nevertheless, BiXT shows that it is possible to achieve $82.0\%$ on ImageNet using only 15M parameters and no vision-specific internal components.

### 3.4 SEMANTIC IMAGE SEGMENTATION

We investigate the transferability of our methods onto semantic image segmentation on the ADE20K dataset (Zhou et al., 2017). We follow common practice and integrate BiXT pretrained on ImageNet1K together with SemanticFPN (Kirillov et al., 2019) as decoder, train for 80k iterations with learning rate $6e^{-5}$ and weight decay $0.01$ following El-Nouby et al. (2021) and others. We choose a batch size of 32 due to the efficiency of our model on the $512^2$ images. Our vanilla BiXT performs competitively against other methods with similar FLOP counts, while the more vision-specific version BiXT+LPI is on par with even the improved PvTv2 and outperforms the others (Table 3).

**Criticism on decoders & a potential alternative.** Decoders like SemFPN were originally introduced for CNN-like architectures and use feature maps at multiple resolutions. Non-hierarchical Transformer architectures like BiXT thus need to downsample and up-convolve their feature maps at various stages – raising the question how this affects performance and to which extent results are caused by backbone, decoder and the compatibility of the two. To provide insights unaffected by these potential influences, we take inspiration from the recently published DINOv2 (Oquab et al., 2023) and simply use a linear layer to directly predict a segmentation map at feature resolution from the last layer's tokens, which we then upsample using bilinear interpolation. Interestingly, our naive approach even outperforms our SemFPN variant, indicating that more research into the suitability of these decoders with non-hierarchical architectures might be needed.

Table 2: **Classification on ImageNet1K using 'few-FLOP' Transformers.** Note that we focus here on efficient models in the low FLOP and/or parameter regime. Perceiver architectures are included as contrast to our bi-directional attention. All methods have been trained on input resolutions of $224^2$, and ↑384 further fine-tuned on $384^2$. Note that different models may have received a different optimization effort. *result reproduced as not reported in original work.

| Architecture | FLOPs | #Param | Acc. |
|---|---|---|---|
| *'Generalists' – no tokenizer, no vision-specific internals* | | | |
| Perceiver (Jaegle et al., 2021) | 707G | 45M | 78.0 |
| Perceiver v2 (Jaegle et al., 2022) | 404G | 42M | 78.6 |
| Perceiver-IO (Jaegle et al., 2022) | 407G | 48M | 79.0 |
| *'Vanillas' – tokenizer, but no vision-specific internals* | | | |
| Perceiver v2 (conv) (Jaegle et al., 2022) | 367G | 42M | 77.4 |
| Perceiver-IO (conv) (Jaegle et al., 2022) | 369G | 49M | 82.1 |
| DeiT-Ti/16 (Touvron et al., 2021) | 1.3G | 6M | 72.2 |
| DeiT3-Ti/16* (Touvron et al., 2022) | 1.3G | 6M | 75.4 |
| BiXT-Ti/16 | 1.7G | 15M | 79.1 |
| *Vision-specific derivatives, incl. multi-scale* | | | |
| PiT-Ti (Heo et al., 2021) | 0.7G | 5M | 73.0 |
| PiT-XS (Heo et al., 2021) | 1.4G | 11M | 78.1 |
| ViL-Ti-APE (Zhang et al., 2021) | 1.3G | 7M | 76.3 |
| ViL-Ti-RPB (Zhang et al., 2021) | 1.3G | 7M | 76.7 |
| PVTv1-Ti (Wang et al., 2021) | 1.9G | 13M | 75.1 |
| PVTv2-B1 (Wang et al., 2022) | 2.1G | 13M | 78.7 |
| BiFormer (Zhu et al., 2023) | 2.2G | 13M | 81.4 |
| XCiT-T12 (El-Nouby et al., 2021) | 1.2G | 7M | 77.1 |
| XCiT-T24 (El-Nouby et al., 2021) | 2.3G | 12M | 79.4 |
| BiXT-Ti/16 (+LPI from XCiT) | 1.7G | 15M | 79.9 |
| *Going finer w/ BiXT – smaller patches, larger images* | | | |
| BiXT-Ti/8 | 4.7G | 15M | 80.8 |
| BiXT-Ti/4 | 16.8G | 15M | 81.2 |
| BiXT-Ti/16 ↑384 | 3.6G | 15M | 81.0 |
| BiXT-Ti/8 ↑384 | 12.5G | 15M | 81.8 |
| BiXT-Ti/4 ↑384 | 48.1G | 15M | 82.0 |

## 3.5 Beyond 2d grid-based image data: Point cloud part segmentation

Table 3: **Semantic Segmentation on ADE20K.** We again focus here on efficient models in the low FLOP and/or parameter regime. All methods have been trained on $512^2$ images.

| Backbone | FLOPs | #Param | mIoU. |
|---|---|---|---|
| *Using the Semantic FPN decoder (Kirillov et al., 2019)* | | | |
| PVTv2-B0 (Wang et al., 2022) | 25.0G | 8M | 37.2 |
| ResNet18 (He et al., 2016) | 32.2G | 16M | 32.9 |
| PVTv1-Ti (Wang et al., 2021) | 33.2G | 17M | 35.7 |
| PVTv2-B1 (Wang et al., 2022) | 34.2G | 18M | 42.5 |
| XCiT-T12 (El-Nouby et al., 2021) | – | 8M | 38.1 |
| BiXT-Ti/16 | 31.8G | 19M | 37.9 |
| BiXT-Ti/16 (+LPI from XCiT) | 32.4G | 19M | 42.4 |
| *Simple linear predictor* | | | |
| BiXT-Ti/16 | 6.4G | 15M | 38.4 |
| BiXT-Ti/8 | 23.2G | 15M | 40.8 |

Since BiXT provides a similar generality to Perceiver regarding its input data structure but additionally allows the use of the dense, local token information, we run experiments to determine its suitability regarding the segmentation of sub-parts of a point cloud – commonly referred to as *point cloud part segmentation* – on the ShapeNetPart (Yi et al., 2016). With recent methods like PointMLP (Ma et al., 2022) achieving a class mIoU of up to $84.6\%$ (instance mIoU of $86.1\%$), the naive application of BiXT with a linear classifier directly applied to the last layer's tokens achieves a class mIoU of $83.5\%$ (instance mIoU of $85.2\%$). Note, however, that methods in this space are usually highly specialized encoder-decoder structures. As in previous experiments, including a modality-specific token-refinement ('geometric affine grouping') and passing the encoded information to PointMLP's decoder (Ma et al., 2022) lets BiXT obtain a highly competitive class mIoU of $84.7\%$ (instance mIoU $86.0\%$) – as always trading off performance and generality.

Table 4: **Point cloud part segmentation on ShapeNetPart** (Yi et al., 2016). Reported are the class IoU and instance IoU for BiXT and PointMLP (Ma et al., 2022). Note that we only compare to PointMLP due to investigating the use of their grouping module and decoder within BiXT.

| Method | Cls. mIoU | Inst. mIoU | aero-plane | bag | cap | car | chair | ear-phone | guitar | knife | lamp | laptop | motor-bike | mug | pistol | rocket | skate-board | table |
|---|---|---|---|---|---|---|---|---|---|---|---|---|---|---|---|---|---|---|
| PointNet | 80.4 | 83.7 | 83.4 | 78.7 | 82.5 | 74.9 | 89.6 | 73.0 | 91.5 | 85.9 | 80.8 | 95.3 | 65.2 | 93.0 | 81.2 | 57.9 | 72.8 | 80.6 |
| PointMLP | 84.6 | 86.1 | 83.5 | 83.4 | 87.5 | 80.5 | 90.3 | 78.2 | 92.2 | 88.1 | 82.6 | 96.2 | 77.5 | 95.8 | 85.4 | 64.6 | 83.3 | 84.3 |
| **BiXT** (naïve) | 83.5 | 85.1 | 83.9 | 81.4 | 91.5 | 79.0 | 89.5 | 76.2 | 91.9 | 87.3 | 79.3 | 95.8 | 73.1 | 95.0 | 84.2 | 63.7 | 80.4 | 83.5 |
| **BiXT** (EncDec) | 84.7 | 86.0 | 84.4 | 82.7 | 86.3 | 80.9 | 90.2 | 80.1 | 92.1 | 87.8 | 82.3 | 95.9 | 78.1 | 95.9 | 84.9 | 67.0 | 82.4 | 83.9 |

## 3.6 Scaling up – Number of latents vs. latent dimension

The majority of this paper is concerned with tiny efficient models – however, it is interesting to see whether our models follow previous Transformers in terms of scaling. BiXT offers an additional degree of freedom in the number of latents. We therefore provide some insights into BiXT's ImageNet1K performance for $32, 64$ and $128$ latents paired with 'tiny' ($D = 192$) and 'small' ($D = 384$) configurations (Table 5). As expected, accuracy increases with both larger embedding dimension and number of latents – while it is worth noting that increasing the number of latents scales quadratically due to the self-attention based latent refinement. Note that while we chose not to run excessive hyperparameter optimisation and refrain from translating to very large architectures due to the large computational requirements involved, we did not observe any signs why BiXT should not behave like other Transformer architectures in terms of scaling and performance. We therefore anticipate to see similar tendencies for larger architectural variants, following the trend reported in related attention-based architectures, but leave this to future work.

## 3.7 Limitations

Our results obtained from the investigation of iterative vs. bi-directional attention in Section 3.2 clearly indicate that bi-directional attention is advantageous in a number of settings with 'medium' to 'large' input sequence length, including finely tokenized images of up to 7056 tokens. However, it is worth noting that by simultaneously refining the tokens alongside the latents, BiXT does not decouple the model's depth from the input and output, unlike Perceiver models (Jaegle et al., 2021). Therefore, very deep BiXT variants might face difficulties in settings of attending to extremely long sequences like raw pixels of large images paired with limited compute and memory. However, we suspect most such scenarios to benefit from some form of preprocessing via a modality-specific input tokenizer,

Table 5: **Scaling up.** Top-1 classification results on ImageNet1k for varying numbers of latents and embedding dimension. All models have been trained for 300 epochs.

| #Latents | *tiny* (D=192) | | | | *small* (D=384) | | | |
|---|---|---|---|---|---|---|---|---|
| | Acc. (%) | FLOPs | Mem | #Param | Acc. (%) | FLOPs | Mem | #Param |
| 32 | 76.68 | 1.30G | 5.53M | 15.11M | 80.60 | 5.02G | 11.07M | 59.57M |
| 64 | 78.13 | 1.68G | 7.23M | 15.11M | 81.20 | 6.43G | 14.53M | 59.59M |
| 128 | 78.80 | 2.47G | 10.95M | 15.13M | 81.87 | 9.32G | 21.90M | 59.61M |

similar to the input-adapter-based concept used in Jaegle et al. (2022) – shifting most applications again into domains where BiXT performs effectively and efficiently.

## 4 RELATED WORK

The introduction of Transformers by Vaswani et al. (2017) has helped *self-attention* to significantly gain in popularity, despite its caveat of scaling quadratically in computational time and memory with input length. Their flexibility regarding input modality and success in Natural Language Processing (NLP) (Devlin et al., 2019) and Computer Vision (CV) (Dosovitskiy et al., 2021; Touvron et al., 2021; 2022) has recently attracted interest in more efficient versions, prompting a series of works.

Approximating the attention matrix via low-rank factorization has been employed across NLP (Katharopoulos et al., 2020; Wang et al., 2020; Song et al., 2021), CV (Chen et al., 2018; Zhu et al., 2019; Li et al., 2019) and others (Choromanski et al., 2021), essentially avoiding the explicit computation through associativity, estimating a set of bases or using sampling – usually at the expense of performance. Others proposed to use tensor formulations (Ma et al., 2019; Babiloni et al., 2020) or exploit the input data structure (Parmar et al., 2018; Ho et al., 2019; Qiu et al., 2020; El-Nouby et al., 2021) under the umbrella of sparsity, however limiting their use to only one specific input modality.

The line of work closest related to ours are 'memory-based approaches' which employ some form of global memory to allow indirect interaction between local tokens. Beltagy et al. (2020) propose to compose various local windowed patterns (sliding, dilated) with global attention on few 'pre-selected' and task-specific input locations for NLP tasks, while its vision derivative (Zhang et al., 2021) provides global memory as tokens within a vision-pyramid architecture and employs four different pairwise attention operations combined with several sets of global tokens that are discarded at certain stages, introducing rather high architectural complexity. Ainslie et al. (2020) follow a similar strategy but additionally investigate the encoding of structured NLP inputs, whereas Zaheer et al. (2020) propose a hand-crafted mix of random, window and global attention to sparsify and thus reduce attention complexity. While their idea of indirect local token communication via a shared global memory aligns with ours, BiXT realizes this goal in a much simpler and modality-independent manner when compared to the mix of highly modality-specific components, attention patterns and strategies involved in these works. Preserving generality w.r.t. to input, Lee et al. (2019) use a set of learnable 'inducing points' via cross-attention to query input data, while the recent Perceiver architectures (Jaegle et al., 2021; 2022) similarly a fixed set of latents to query input data – yet none offers the efficient simultaneous refinement of latents and tokens realized in our BiXT.

## 5 CONCLUSION

In this paper, we have presented a novel bi-directional cross-attention Transformer architecture (BiXT) for which computational cost and memory consumption scale linearly with input size, by leveraging a naturally emerging symmetry in two-way cross-attention that aligns with common intuition and has been empirically validated in this work. By allowing the 'what' (latent variables) and 'where' (input tokens) to attend to each other simultaneously and develop alongside throughout the architectural stages, BiXT combines Perceiver's linear scaling with full Transformer architectures' high performance in a *best-of-both-worlds* approach. The ability to process longer sequences paired with the ease to integrate further domain-specific token refinement modules helps BiXT to perform competitively on a number of point cloud and vision tasks, and to outperform comparable methods on ImageNet1K in the low-FLOP regime, both in its 'vanilla' and 'vision-specific' form.

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

## A  BIXT – GENERAL ASPECTS AND INSIGHTS

### A.1  CODE AND REPRODUCIBILITY

We implemented our models in PyTorch (Paszke et al., 2019) using the timm library, and will release all code and pretrained models. We further made use of the mmsegmentation library (Contributors, 2020) for the semantic segmentation experiments. Point cloud experiments were built on the publicly released code base from (Ma et al., 2022).

### A.2 COMPLEXITY ANALYSIS.

The complexity of BiXT is dominated by the bi-directional cross-attention, in particular by a) the matrix multiplication to compute the similarity matrix and b) the two matrix multiplications to compute the refined outputs. Using the previously specified embedding dimension $D$, $N$ tokens and $M$ latent vectors, multiplication a) involves matrices of shape $M \times D, D \times N$ with result $M \times N$, and the two multiplications b) involve matrices of shape $M \times N, N \times D$ with result $M \times D$ and $N \times M, M \times D$ with result $N \times D$. The overall complexity per layer is thus $\mathcal{O}(MND) = \mathcal{O}(N)$ and linear in the size of the input $N$.

### A.3 TYPES OF ATTENTION – ADDITIONAL RESULTS

An extended list of the results stated in Section 3.2 are presented in Table A1. Note that we performed an individual sweep over a set of learning rates for each individual architecture – usually starting at $4e^{-3}$ and lowering until stable training occurred. We then used these results to pick the best 5 architectural variants and training schemes, and ran them for an additional 2 random seeds. Note that all architectural variants, including BiXT and the sequential one have only been run in this setup for a total of maximum 3 runs, and no cherry-picking of results occurred for either of the architectures. Note that we have also tried stepped schedulers with the schedule proposed in the original Perceiver paper (Jaegle et al., 2021), but resorted back to using the cosine since it showed equal or superior results.

To contrast the sequential attention to our default BiXT with *12 layers* (d12) on a matching FLOP level, the sequential version uses *only 11 layers* (d11) due to its higher complexity per layer. This is due to the fact that our bi-directional cross-attention only requires 4 instead of 6 projection matrices ($2 \times [R, V]$ vs. $2 \times [Q, K, V]$) and only computes the attention matrix once (instead of twice). The hereby saved FLOPs (as well as parameters and memory) can then be spent on additional layers, further improving results by another $\sim 1.2\%$. Architectures with 1 more layer each show the same trend.

In other words, by holding FLOP and/or memory requirements constant, we consistently observe a net benefit with our bi-directional attention in terms of accuracy throughout our experiments. We empirically found that it additionally improved robustness/consistency across different parameter initializations (seeds), which can be seen by the smaller standard deviations of the bi-directional variants.

## B FURTHER DETAILS ON IMAGENET1K EXPERIMENTS

This section outlines further details regarding our image classification experiments conducted on the ImageNet dataset (Russakovsky et al., 2015).

### B.1 MODEL CONFIGURATIONS AND TRAINING DETAILS

Hyperparameter choice for the default ImageNet experiments: BiXT with 64 latents, 12 layers – learning rate $2.5e^{-3}$, weight decay $0.05$ and lambc optimizer, as well as cosine learning rate scheduler; stochastic dropout on self-attention and cross-attention $0.05$ for all tiny models. Apart from these, we directly apply the augmentation and training proposed by Touvron et al. (2022). Our models have been trained between 300 and 800 epochs on one or several A100 GPUs. Note that we did not conduct an extensive hyperparameter search, and expect results to potentially improve if done so. Finetuning was performed for 30 epochs using an initial learning rate of $4e^{-5}$ with cosine decline.

For the experiments where we investigate the scaling properties of 'small' architectures, we follow the same strategy but increase the stochastic path drop rate for self- and cross-attention to between $0.1 - 0.2$ to prevent overfitting of the architectures.

### B.2 ABLATING PATCH SIZE FOR FIXED SEQUENCE LENGTH IN IMAGE CLASSIFICATION

In this section, we investigate whether lowering the patch size to increase the resolution of the resulting feature maps is actually the most-suited way – or whether simply reducing the stride and

Table A1: **Architectural variants using iterative attention & cross-attention parameter sharing.** Classification accuracy on the ImageNet1K dataset for varying types of attention. All architectures use 64 latent vectors and have been trained for 120 epochs with hyperparameters individually optimized. Cross-attention parameter sharing schemes: [†]indicates sharing of all, [‡]of all but the 1st layer's cross-attention parameters. Architectural configurations noted in brackets. Three randomly seeded runs were performed for the 'best' architectures (judged by their performance on seed = 42), and mean and (unbiased) standard deviation are reported. One randomly seeded run reported for all other architectures.

| Attention type | Acc.@1 (%) | Acc.@5 (%) | FLOPs | Mem. | #Param |
|---|---|---|---|---|---|
| Iterative[†] (sa5-d8) | 57.51 | 80.61 | 1.58G | 7.17M | 18.61M |
| Iterative[†] (sa6-d7) | 58.86 | 81.53 | 1.59G | 7.23M | 19.50M |
| Iterative[†] (sa6-d8) | $60.61 \pm 1.11$ | $82.75 \pm 0.68$ | 1.82G | 8.25M | 22.16M |
| Iterative[†] (sa4-d12) | $56.03 \pm 1.02$ | $79.38 \pm 0.80$ | 1.99G | 9.10M | 22.16M |
| Iterative[†] (sa1-d22) | 56.09 | 79.36 | 1.64G | 7.70M | 11.04M |
| Iterative[†] (sa1-d24) | $55.92 \pm 0.67$ | $79.33 \pm 0.52$ | 1.79G | 8.39M | 11.93M |
| Iterative[‡] (sa5-d8) | $58.26 \pm 2.34$ | $81.02 \pm 1.76$ | 1.58G | 7.17M | 19.05M |
| Iterative[‡] (sa6-d7) | $54.94 \pm 5.96$ | $78.39 \pm 4.69$ | 1.59G | 7.23M | 19.94M |
| Iterative[‡] (sa6-d8) | 58.23 | 80.95 | 1.82G | 8.25M | 22.61M |
| Iterative[‡] (sa4-d12) | 56.35 | 79.64 | 1.99G | 9.10M | 22.61M |
| Sequential (2-way, d11) | $71.64 \pm 0.45$ | $90.39 \pm 0.41$ | 1.66G | 7.52M | 14.60M |
| Sequential (2-way, d12) | $72.72 \pm 0.76$ | $90.95 \pm 0.44$ | 1.81G | 8.19M | 15.92M |
| Bi-Directional (d12) | $72.48 \pm 0.31$ | $90.83 \pm 0.19$ | 1.68G | 7.23M | 15.11M |
| Bi-Directional (d13) | $73.61 \pm 0.34$ | $91.42 \pm 0.19$ | 1.82G | 7.89M | 16.38M |

Table A2: **Varying patch sizes for fixed sequence lengths.** ImageNet1k classification results for varying patch sizes are presented for three fixed sequence lengths (realised via stride). All models have been trained for 300 epochs using the same (default) hyperparameters and input images of size $224 \times 224$. Best results for each sequence length is highlighted in bold.

| Seq. length | 196 ($14 \times 14$) | | 784 ($28 \times 28$) | | | 3136 ($56 \times 56$) | | |
|---|---|---|---|---|---|---|---|---|
| Patch size | $32 \times 32$ | $16 \times 16$ | $32 \times 32$ | $16 \times 16$ | $8 \times 8$ | $16 \times 16$ | $8 \times 8$ | $4 \times 4$ |
| Acc. (%) | 77.50 | **78.13** | 79.90 | **79.92** | 79.36 | **80.95** | 80.75 | 79.56 |
| FLOPs | 1.77G | 1.68G | 5.05G | 4.71G | 4.62G | 16.81G | 16.46G | 16.38G |
| Mem | 7.27M | 7.23M | 20.25M | 20.25M | 20.25M | 72.18M | 72.18M | 72.18M |
| #Param | 15.56M | 15.11M | 15.56M | 15.12M | 15.01M | 15.12M | 15.01M | 14.98M |

thus creating tokens that originate from overlapping patches yield better results. Our experiments on image classification using the ImageNet1k (Russakovsky et al., 2015) dataset with models using varying patch sizes and strides to keep the sequence lengths fixed show that the originally introduced and commonly used patch size of $16 \times 16$ pixels seems to be a good fit when using no overlapping patches (Table A2). Interestingly, we find that even when we increase the feature resolution and thus choose smaller strides, a patch size of $16 \times 16$ still yields best results across our experiments. One potential reason is that patch boundaries are randomly chosen and objects in images do naturally not match these boundaries, so that information has to be exchanged – whereas slight overlaps might ease this to some extent. Another potential reason for this behaviour is that significantly decreasing the patch size reduces the input information per patch, with an $8^2$ RGB patch having a total of 192 channels, exactly matching the tiny embedding dimension. Lower patches however would create a significant null space, which might be an additional reason for better performance when using larger patches.

### B.3 TOKEN REFINEMENT VIA LOCAL PATCH INTERACTION (XCIT)

We integrate the 'LPI' module from El-Nouby et al. (2021) together with their convolutional tokenizer for our vision-specific experiments. The LPI module consists of two depth-wise convolutional layers (3x3) with Batch Normalization and a GELU non-linearity in between. For further details, please refer to the original paper.

## C FURTHER DETAILS ON POINT CLOUD EXPERIMENTS

Note that we do not use any voting strategy or other multi-scale augmentation and simply follow the training regime of PointMLP (Ma et al., 2022) for most of our experiments. We use a standard BiXT architecture for the 'naive' point cloud experiments as well as the ones using simple grouping – and reduce our architecture to 4 layers when using the decoder for part segmentation and the hierarchical approach for shape classification – paired with 32 and 24 neighbours, respectively (default values used in other works like PointMLP).

## D VISUALIZATION OF LATENT-TOKEN ATTENTION

To provide some additional qualitative insights into the bi-directional attention that is cast within BiXT, we provide three sets of attention maps overlaid onto the input image:

- Figure A1: The attention maps of the four latent vectors presented in Figure 1(d) for all layers throughout the BiXT tiny architecture (layer 1, top-left to layer 12, bottom-right).
- Figure A2 The attention maps of all latent vectors (64 in this case) for the final layer of our BiXT tiny architecture.
- Figure A3 The attention maps of all latent vectors (64 in this case) for the second-last layer of our BiXT tiny architecture.

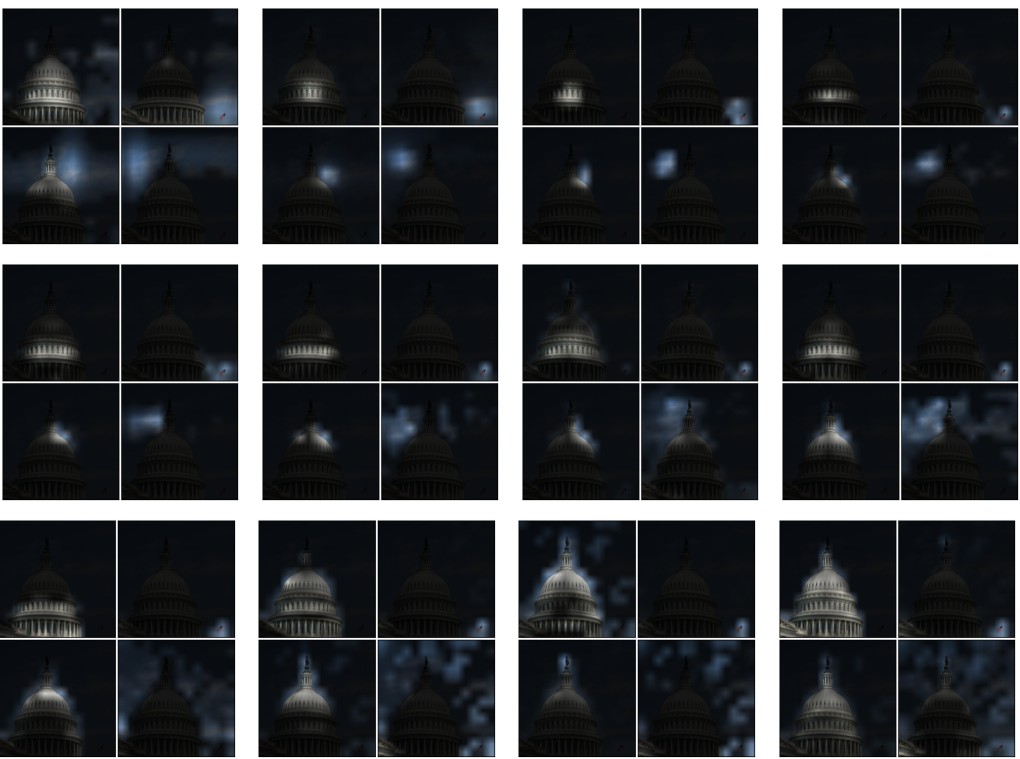

Figure A1: **Attention across layers.** Bi-directional attention maps for the four selected tokens presented in Figure 1(d) across all layers: Starting with first layer on top left, ending with last layer (layer 12) on the bottom right. Displayed are the mean attention maps averaged across the 3 heads of BiXT tiny with 64 latents.

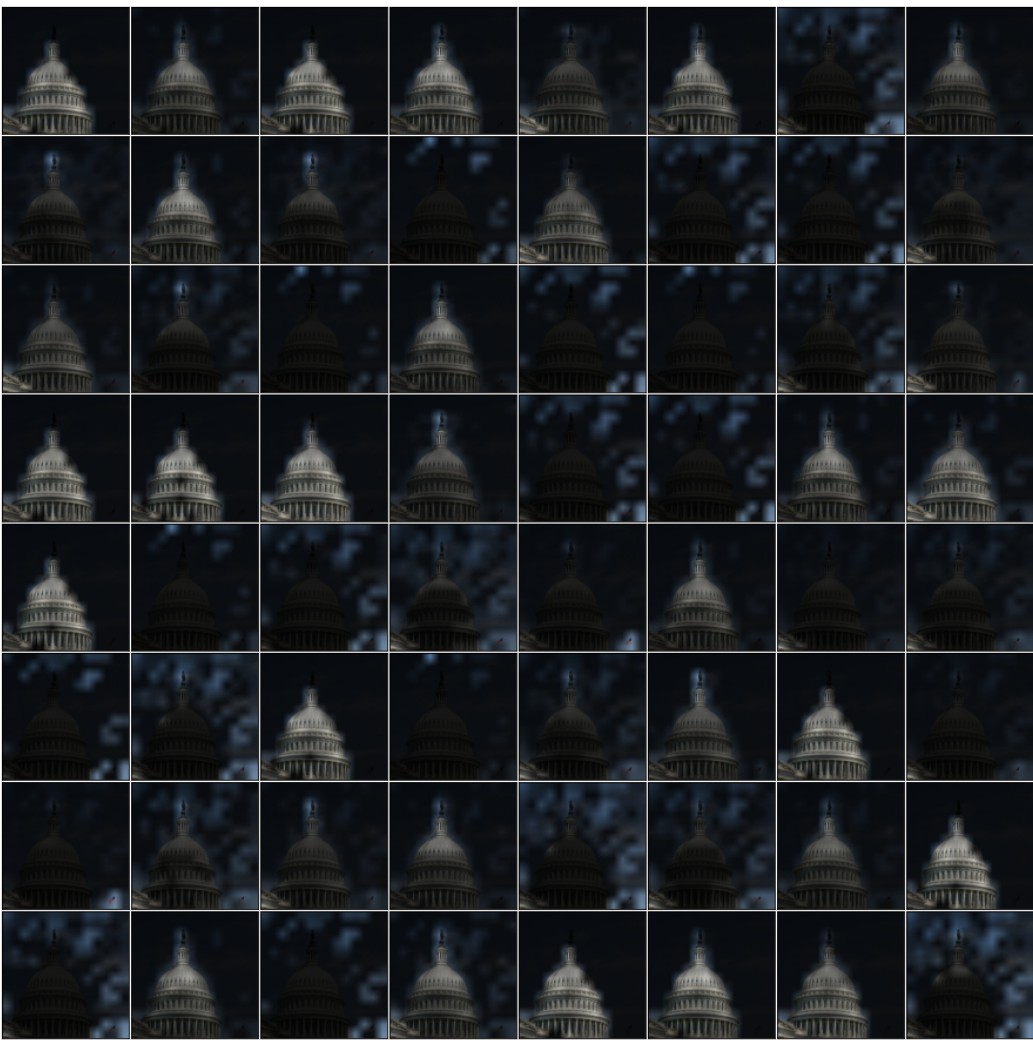

Figure A2: **Attention maps of final layer.** Bi-directional cross-attention maps of all 64 latent vectors of the final layer (layer 12) of our BiXT tiny architecture.

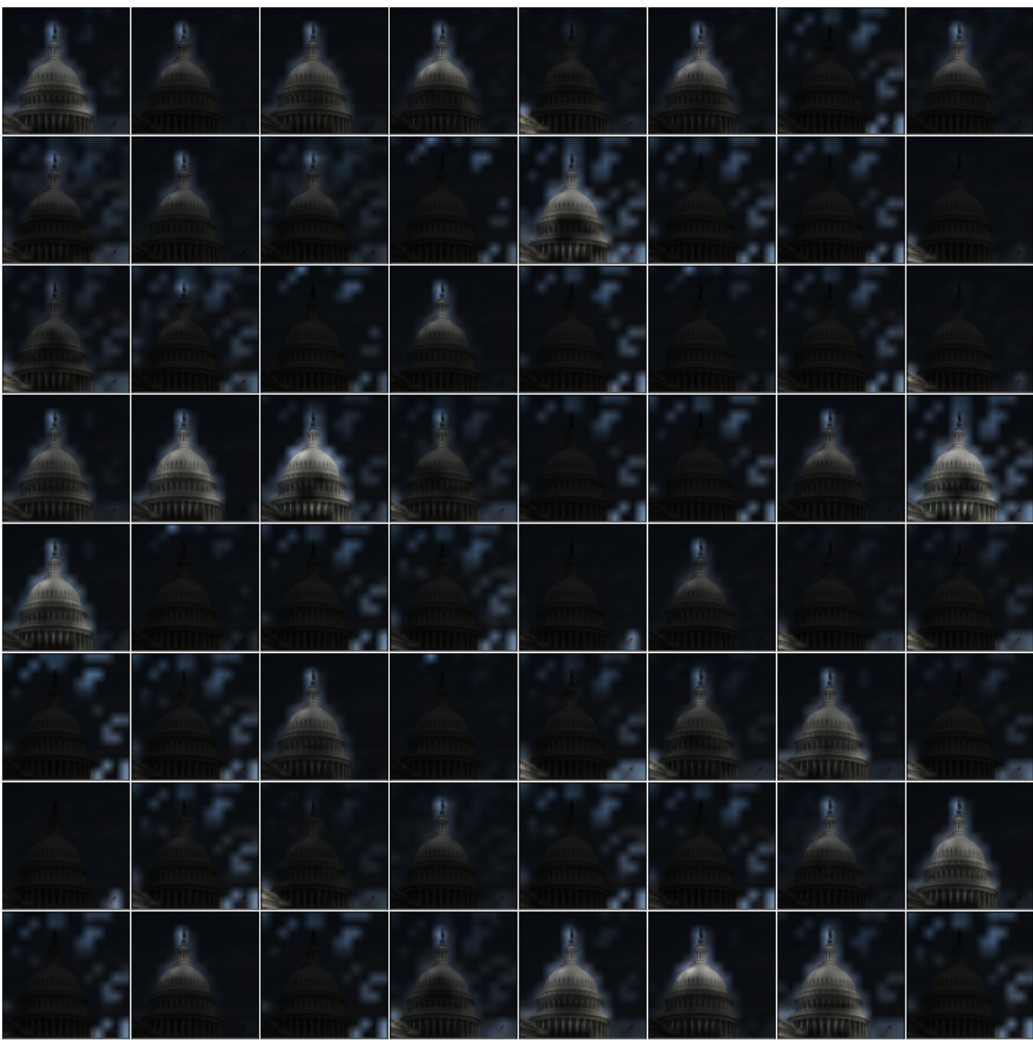

Figure A3: **Attention maps of penultimate layer.** Bi-directional cross-attention maps of all 64 latent vectors of the second-last layer (layer 11) of our BiXT tiny architecture.

