# OpenReview forum: "BiXT: Perceiving Longer Sequences With Bi-Directional Cross-Attention Transformers"
_ICLR.cc/2024/Conference — Submitted to ICLR 2024_

### Official Review · Reviewer_bDbJ · 2023-10-29

**Soundness:** 2 fair
**Presentation:** 2 fair
**Contribution:** 2 fair
**Rating:** 1
**Confidence:** 5

**Summary:**

This paper proposes a Bi-Directional cross-attention to model the interactions of the visual tokens. Experiments on ImageNet1K and ShapeNetPart are performed to evaluate the effectiveness of the proposed method.

**Strengths:**

1. This paper is well-written and easy to follow.

**Weaknesses:**

1. The novelty is limited. Cross-attention has been widely used for years and the proposed method is simply some combination of cross-attention operation.
2. The experimental results are not very impressive. The accuracy on ImageNet is only 82.0, which is not competitive.

**Questions:**

see the weaknesses.

---

> ### Author Response · Authors · 2023-11-18
> **Response to Reviewer bDbJ**
>
> We thank you for your time put into reviewing our work. However, we have to respectfully disagree with the two stated weaknesses:
>
> **[W1]: Limited novelty, Cross-Attention widely used**
> > *The novelty is limited. Cross-attention has been widely used for years and the proposed method is simply some combination of cross-attention operation.*
>
> We'd like to clarify that we do not claim to introduce cross-attention, nor do we simply combine CAs. While we do analyze a naïve version of sequential CA in our paper as comparison, one of the core contributions of our work is the introduction of a *novel efficient bi-directional cross-attention method* which is built upon the *observation of symmetric tendencies* that emerge in the attention patterns. This combined with our BiXT architecture scales linearly with the input sequence length, is generally applicable to a variety of input modalities, and additionally saves compute, memory and parameters.
>
> ---
> **[W2]: Experimental results / not beating SOTA on ImageNet**.
> > *The experimental results are not very impressive. The accuracy on ImageNet is only 82.0, which is not competitive.*
>
> Note that we achieve the 82% with our **general** architecture that is **applicable across different input modalities** (not only images) and scales linearly w.r.t. the input sequence length. Our results therefore need to be *interpreted in the context of other such methods* like the recent Perceiver~IO (2022) models, which however perform either worse and/or require significantly more compute as we demonstrate in Table 2:
> | Method | Top-1 Acc | FLOPs |
> |:----|:----:|:-----:|
> Perceiver-v2 | 77.4% - 78.6% | 367-404G
> Perceiver-IO |  79.0% - 82.1% | 369-407G
> BiXT (ours) |  79.1% - 82.0% | **1.7- 48G**
>
> ---
> We hope this sheds some new light on your interpretation of our work. Please do let us know if we can provide any additional details or clarify potentially misleading points, and we encourage you to further elaborate in context of this new information.

---

> ### Author Response · Authors · 2023-11-23
> **Gentle reminder**
>
> Dear Reviewer bDbJ,
>
> We hope this message finds you well. As the deadline for the discussion phase is approaching, we wanted to check if you have any remaining questions that we could help clarify.
>
> Warm regards,
> The Authors

---

### Official Review · Reviewer_qx3u · 2023-10-30

**Soundness:** 3 good
**Presentation:** 3 good
**Contribution:** 3 good
**Rating:** 6
**Confidence:** 4

**Summary:**

This research paper provides an enhancement to the Perceiver architecture that employs latent queries for the distillation of input tokens. The main novelty introduced is a bidirectional cross-attention module aimed at reducing computational demands. The authors analyze an architecture that iteratively stacks query-to-token and token-to-query cross-attention modules and find symmetry between these two attention values, suggesting that these two iteratively stacked modules can be merged into one. Therefore, a new bidirectional transformer architecture that only scales linearly with the input tokens as a means of dealing with general modal input data. This replacement results in a reduction of computational cost by approximately one-third, compared to iterative stacking cross-attentions, while also achieving higher accuracies.

The improved method demonstrates an impressive 82.0% accuracy for classification tasks on ImageNet-1K using compact models. These models require only a small fraction of the FLOPS compared to the original Perceiver. The paper also provides verification tests on more generalized input modalities, reinforcing the versatility and effectiveness of the proposed enhancements to the Perceiver architecture.

**Strengths:**

1. The idea of introducing bidirectional attention to replace the iterative stacking cross attention is both innovative and simple.
2. The way that the authors present their idea is also appreciated. An analog between the latent queries and "what" queries, and that between the input tokens and the "where" information, is first presented. Then, the symmetry between the "what" and "where" tokens are exemplified to suggest the improvement of the bidirectional attention. Overall, I enjoy reading this paper, and it is easy to follow.
3. The bi-directional cross-attention is effective in reducing the computational cost of the Perceiver architecture and increasing its performance. It achieves the same performance with only a fraction of FLOPS.

**Weaknesses:**

- Despite its effectiveness, the motivation is more from an intuitive analogy of "what" and "where" tokens than a comprehensive theoretical or experimental conclusion. Only an image classification task is presented when analyzing the symmetry between iterative cross attentions between "what" and "where" tokens. There could also exist many others scenarios, where these cross attention value may violate the symmetry property. For example, the “what” tokens would attend to the context background tokens when detecting small object in the image, whereas these attended “where” tokens would more likely to attend to other “what” tokens in the next cross attention. Therefore, it is less convincing to conclude the “symmetry” behavior of the “what” and “where” tokens from a single illustration.


- It is surprising and strange in Table 1(a) that the most performance gain is brought by the naive approach that sequentially stacking two cross attentions with reverse orders by exchanging the query and key positions (+ 11 acc); whereas bidirectional only brings in an additional 0.8 acc. This result seems a bit contradictory with the emphasis of the paper on the bi-directional attention. In this regard, a more important part about the reason of the largely improved performance of the sequential cross attention deserves more detailed analysis. Specifically, this phenomenon would highlight the importance of refining the image tokens instead of fixing them as in Perceiver.

- The FLOPSs and Params reported in Table 1(a) is confusing. The FLOPSs and Params of bi-directional cross attention are even larger than that of sequential cross attention. However,  it is described that the implementation of bidirectional cross attention saves 1/3 parameters compared to naive sequential cross attention. Results in Table 1(a) contradicts this statement.

**Questions:**

See the above.

---

> ### Author Response · Authors · 2023-11-18
> **Response to Reviewer qx3u (1/2)**
>
> Thank you for the comprehensive review and the constructive comments. We address each of them individually in the following.
>
> **[W1]: Symmetry constraint**
> > *Despite its effectiveness, the motivation is more from an intuitive analogy of "what" and "where" tokens than a comprehensive theoretical or experimental conclusion. Only an image classification task is presented when analyzing the symmetry between iterative cross attentions between "what" and "where" tokens. There could also exist many others scenarios, where these cross attention value may violate the symmetry property. For example, the “what” tokens would attend to the context background tokens when detecting small object in the image, whereas these attended “where” tokens would more likely to attend to other “what” tokens in the next cross attention. Therefore, it is less convincing to conclude the “symmetry” behavior of the “what” and “where” tokens from a single illustration.*
>
> We agree with you that there might be cases where complete flexibility could lead to attention maps that differ from a symmetric structure. However, having frequently observed the symmetric tendencies in the sequential cross-attention maps has motivated us to try to leverage this phenomenon to reduce FLOPS, memory and parameters. While this indeed imposes a constraint, we found that the performance was almost unaffected throughout our experiments. When we re-spent the saved compute/memory on additional layers, we were able to consistently obtain a net benefit.
>
> We suspect that the slightly deeper architectures provide our method with enough flexibility to learn any additionally required operations to compensate and realize a more complex information exchange in cases where this might be required (potentially across multiple layers, and/or by using a subset of the latent vectors to facilitate this).
> The exact internal behavior that such shared attention maps yield in terms of information exchange and its structure is a very interesting problem that we consider a promising area for future work.
>
> $\rightarrow$ To provide some further insights for the community, we have added additional visualizations to the appendix of our paper showing the attention maps of all latent vectors (complete set of 64) for the last two layers, as well as the maps of the four latent vectors presented in the paper across all architectural layers (Figures A1-A3, Section D).
> $\rightarrow$ We will also make our code publicly available including our trained models to facilitate future investigations.
>
> *(please also see next response - part 2)*

---

> ### Author Response · Authors · 2023-11-18
> **Response to Reviewer qx3u (2/2)**
>
> **[W2]: Uni-directional & Sequential vs. Bi-directional Attention**
> > *It is surprising and strange in Table 1(a) that the most performance gain is brought by the naive approach that sequentially stacking two cross attentions with reverse orders by exchanging the query and key positions (+ 11 acc); whereas bidirectional only brings in an additional 0.8 acc. This result seems a bit contradictory with the emphasis of the paper on the bi-directional attention. In this regard, a more important part about the reason of the largely improved performance of the sequential cross attention deserves more detailed analysis. Specifically, this phenomenon would highlight the importance of refining the image tokens instead of fixing them as in Perceiver.*
>
> We apologize if this aspect of our paper has been slightly unclear. The main intention of Section 3.2 on "Iterative, sequential or bi-directional" attention is to demonstrate that extending the working memory over Perceiver-like architectures by using both-ways cross-attention can already help to unlock the information bottleneck and significantly improve the results. Using our bi-directional cross-attention built on the observation of emerging symmetric tendencies in the attention maps further boosts the results by improving both efficiency and robustness (w.r.t. hyperparameter changes and across different initializations).
> In other words, saving FLOPs (as well as parameters and memory) via the use of our bi-directional CA means they can be spent on additional layers, further improving results and therefore creating a net benefit in terms of accuracy-to-FLOPs (as well as parameters and memory).
>
> To validate that such behaviour cannot easily be obtained in a Perceiver-style uni-directional manner, we have created and investigated 10 different architectural variants of which the best 5 are presented in Table 1(a) (with the full details reported in the appendix), and contrasted these to a sequential and bi-directional variant of similar FLOP count.
>
> We have now conducted additional experiments for two more architectures to further validate this point: bi-dir with 13 layers and seq-attn with 12 layers (3 seeds each). The results reflect the same trend as the architectures in Table 1 (a), with the bi-directional attention yielding a relative improvement of $\sim 1.22$%  over its sequential counterpart while exhibiting a lower standard deviation across the three randomly seeded training runs:
> | Method | Top-1 Acc | Top-5 Acc | FLOPs | Mem. | #Param |
> |:----|:----:|:-----:|----:|------:|----:|
> | Seq.   (d12) | 72.72±0.76 | 90.95±0.44 | 1.81G | 8.19M | 15.92M |
> | Bi-Dir (d13) | 73.61±0.34 | 91.42±0.19 | 1.82G | 7.89M | 16.38M |
>
> $\rightarrow$ We have included these results into the more detailed overview presented in the appendix (Table~A1).
> If the reviewers consider it helpful to further strengthen our paper, we are happy to include additional results over other variants (e.g. 10, 8 layers) into the final version of our manuscript to further validate this aspect.
>
> ---
> **[W3]: Confusing presentation in Table 1(a)**
> > *The FLOPSs and Params reported in Table 1(a) is confusing. The FLOPSs and Params of bi-directional cross attention are even larger than that of sequential cross attention. However, it is described that the implementation of bidirectional cross attention saves 1/3 parameters compared to naive sequential cross attention. Results in Table 1(a) contradicts this statement.*
>
> The presentation in Table 1(a) has indeed been lacking some clarity. We report architectures that have been matched to a similar FLOP count: between 1.58 - 1.99 for uni-directional attention, and we compare these to the two most similar variants of sequential and bi-directional attention in terms of FLOP count. However, due to the increased efficiency of bi-directional CA, the reported bi-directional model consists of *12 layers*, whereas the sequential model *only has 11 layers*.
>
> $\rightarrow$ We have adapted the methods' names in Table 1(a) and explicitly added the model depth to improve clarity, as well as expanded the details and explanations in the appendix (Section A.3).
>
> ---
> We hope our answers and additional insights have helped to address your questions. Please do let us know if there is any further concern or whether we can provide any further information that is helpful to you.

---

> > ### Comment · Reviewer_qx3u · 2023-11-22
> > **Thanks for addressing my concerns**
> >
> > I would like to thank the authors for clarifying the novelty of this study and the originally missing details in the manuscript. I find my concerns mostly addressed.

---

### Official Review · Reviewer_mPnx · 2023-11-01

**Soundness:** 3 good
**Presentation:** 3 good
**Contribution:** 3 good
**Rating:** 6
**Confidence:** 4

**Summary:**

This paper proposes a bi-directional cross-attention Transformer (BiXT) that can process long sequences efficiently and effectively by using a small set of latent vectors to represent the ‘what’ and input tokens to represent the ‘where’ of the data. At the core of BiXT is the bi-directional cross-attention module that simultaneously refines latent vectors and input tokens. Compared to sequential cross-attention, the bi-directional cross-attention module leverages the symmetry of attention patterns between latent vectors and input tokens to reduce computational cost and memory consumption.  The authors evaluate BiXT on image classification, semantic image segmentation, and point cloud part segmentation.  They show that BiXT outperforms comparable methods in the low-FLOP regime and can easily integrate modality-specific components to improve performance further.

**Strengths:**

1. The proposed bi-directional cross-attention has a simple and neat design
2. Evaluations are conducted on two modalities, i.e., images and point clouds.
3. The paper is well-written, hence easy to follow

**Weaknesses:**

1. Despite the simple and neat design, the strength of the proposed method, bi-directional cross-attention, is unclear. Compared to using two uni-directional cross-attention modules sequentially, the system-level accuracy, FLOPs, and memory requirements are all similar (Table 1 on page 6).

2. Insufficient comparison with some of the latest vision backbones. The methods in image classification (Table 2) and semantic segmentation (Table 3) are somewhat outdated. Many works were proposed to overcome the quadratic complexity of multi-head self-attention, such as MaxViT [1], BiFormer [2], and especially DualViT [3], which has a very similar design to BiXT. The performances of BiXT are not attractive if these approaches are included in comparison. Why are these methods not comparable with BiXT?

3. Lack of experiments with larger models. It is unclear why the comparisons are positioned in a low-FLOP regime (Table 2). BiXT seems not to be specially designed for lightweight models, and the budgets of BiXT-Ti/8 and BiXT-Ti/4 in the final section of Table 2 are sufficient to cover training larger models with more parameters. It may be better to demonstrate the effect of model scaling.

[1] Maxvit: Multi-axis vision transformer, ECCV 2022.
[2] BiFormer: Vision Transformer with Bi-Level Routing Attention, CVPR 2023.
[3] Dual Vision Transformer, TPAMI 2023.

**Questions:**

See the weaknesses part. Overall, I appreciate the simple and neat design of the bi-directional cross-attention. Still, I would like more clarification on its strengths and the experimental settings in the rebuttal. I will raise my rating if these concerns are addressed.

---

> ### Author Response · Authors · 2023-11-18
> **Response to Reviewer mPnx (1/2)**
>
> We thank you for your detailed review and the constructive feedback! We address your comments individually in the following:
>
> **[W1] Sequential vs. bi-directional attention**:
> > *Despite the simple and neat design, the strength of the proposed method, bi-directional cross-attention, is unclear. Compared to using two uni-directional cross-attention modules sequentially, the system-level accuracy, FLOPs, and memory requirements are all similar (Table 1 on page 6).*
>
> This aspect has indeed been lacking some clarity. The architectures reported in Table 1 (a) have been matched to similar FLOP counts: between 1.58 - 1.99 for uni-directional attention, and we contrast them to the two variants of sequential and bi-directional attention that fall into this range (hence the similar FLOPs and memory). The bi-directional architecture using our more efficient attention mechanism has *12 layers* whereas the sequential one has *only 11 layers*.
>
> Our bi-directional CA only requires 4 instead of 6 projection matrices (2x[R,V] vs. 2x[Q,K,V]) and only computes the attention matrix once (instead of twice). The hereby saved FLOPs (as well as parameters and memory) can then be spent on additional layers, further improving results by another $\sim 1.2$% in this configuration.
> In other words, by holding FLOP and/or memory requirements constant, we consistently observed a net benefit with our bi-directional attention in terms of accuracy throughout our experiments. We empirically found that it additionally improved robustness across different parameter initializations (seeds).
>
> To further validate this point, we have conducted additional experiments for two more architectural variants: bi-dir with 13 layers and seq-attn with 12 layers (3 seeds each). These results show the same trend as the two architectures already included in Table 1 (a), with the bi-directional attention outperforming its sequential counterpart while exhibiting a lower standard deviation across the three randomly seeded training runs:
> | Method | Top-1 Acc | Top-5 Acc | FLOPs | Mem. | #Param |
> |:----|:----:|:-----:|----:|------:|----:|
> | Seq.   (d12) | 72.72±0.76 | 90.95±0.44 | 1.81G | 8.19M | 15.92M |
> | Bi-Dir (d13) | 73.61±0.34 | 91.42±0.19 | 1.82G | 7.89M | 16.38M |
>
>
> $\rightarrow$ We have included the results and additional discussion into the detailed overview presented in the appendix of our revised version (Section A.3).
> $\rightarrow$ We have additionally updated the methods' names in Table 1(a) to clearly represent the difference in model depth (and explain the similar FLOP and memory counts).
>
> ---
>
> **[W2-1]: Comparison to recent vision-specific works -- Rel. works**
> > *Insufficient comparison with some of the latest vision backbones. The methods in image classification (Table 2) and semantic segmentation (Table 3) are somewhat outdated. Many works were proposed to overcome the quadratic complexity of multi-head self-attention, such as MaxViT [1], BiFormer [2], and especially DualViT [3], which has a very similar design to BiXT.*
>
> We thank you for drawing our attention to these very interesting works, especially [2] and [3] which we will add to our manuscript ([1] is already included in related work).
>
> Brief discussion of differences to [2] & [3]:
> DualViT's [3] *dual block* used in the early layers of their architecture does indeed show similarity to the na\"ive solution of sequential cross-attention, but is distinctly different from our bi-directional approach as it does not leverage any symmetry. Importantly, their multi-stage pyramidal vision-only architecture uses a large number of `merging blocks/layers' (between 9 - 24) which cast full self-attention over the concatenated sequence of latents and tokens. This prevents linear scaling and also introduces a shared embedding space of latent vectors and tokens through the use of the same key-query-value projection matrices -- whereas our architecture keeps those separate (aligned with the presented 'what' and 'where' analogy and the level of information they represent) and scales linearly with respect to the input length.
>
> Biformer [2] similarly is a pyramidal vision-only approach that reduces the computational complexity through routing information between selected tokens via a directed graph, thus achieving sparsity to skip computation of certain regions that are deemed irrelevant. While this is a very neat way of dynamically reducing complexity, it is distinctly different from our approach and does not achieve true linear scaling.
>
> $\rightarrow$ We will include both works into the related works section of our revised paper, with additional more in-depth discussion in the appendix. (please also see the following response - part2)

---

> ### Author Response · Authors · 2023-11-18
> **Response to Reviewer mPnx (2/2)**
>
> **[W2-2]: Comparison to recent vision-specific works -- Performance**
> > *The performances of BiXT are not attractive if these approaches are included in comparison. Why are these methods not comparable with BiXT?*
>
> We have included the recent BiFormer-T [2] with its 2.2 GFLOPs and 81.4% in our revised manuscript as it does indeed fall into (or close to) our intended FLOP regime. In the comparison of low-FLOP Transformer methods in Table 2, we had decided on a cut-off at 2.0 GFLOPS. The only two included methods slightly above this, i.e. XCiT-T24 with 2.3 GFLOPs and PVTv2-B1 with 2.1 GFLOPs, had been included due to them being improvements/modifications of previous versions of these same architectures (XCiT-T12 and PVTv1).   [1]'s and [3]'s smallest models require however more than double and do thus not fall into this range, neither does any of these for semantic segmentation.
>
> However, we would like to clarify that our intent of this comparison in Table 2 is to demonstrate that the core of our method is mostly orthogonal to domain-specific approaches and that many of their techniques can be used to easily extend our architecture to further improve results while trading off generality. The important comparison is hence *'general BiXT'* vs. *'BiXT + LPI'* (and potentially vs. XCiT), where LPI was simply chosen to provide *one example* of such a possible extension (of which there are many, including pyramidal structures). We see a more dense exploration of such domain-specific variants and interactions as an interesting future research direction.
>
> $\rightarrow$ We have updated Table 2 to represent [2] in our revised version and will modify the wording of the describing paragraph accordingly to better outline our intent.
>
> ---
> **[W3]: Larger models & reasons for conducted experiments**
> > *Lack of experiments with larger models. It is unclear why the comparisons are positioned in a low-FLOP regime (Table 2). BiXT seems not to be specially designed for lightweight models, and the budgets of BiXT-Ti/8 and BiXT-Ti/4 in the final section of Table 2 are sufficient to cover training larger models with more parameters. It may be better to demonstrate the effect of model scaling.*
>
> There are two reasons for our focus on a low-FLOP regime:
>  1) The limited computational budget available to us and many other researchers, which only allows us to run a few larger experiments, and
>  2) The aspect that our bi-directional cross-attention is focused on scaling linearly as well as requiring fewer FLOPs, memory and parameters than a naïve sequential realization.
>
> Since one of the core attributes of BiXT is its linear scaling w.r.t. the input sequence length, we chose to spend our available budget to showcase the effect & benefit which BiXT's ability to process longer sequences yields. Note that since the investigated architecture stays the same and only more tokens are passed as input, we were able to directly use the exact same training hyperparameters as we used in the other experiments.
>
> While it would indeed be interesting to additionally analyze large models, we'd like to note that this would require a substantial number of additional large experiments. Even though such models might at first appear to require similar compute, the actually required computational budget not only encompasses the training runs but also the hyperparameter search.
> The importance of well-chosen hyperparameters and augmentation strategies grows significantly with model size, as can be seen in the literature (e.g. in the transition from ViT$\rightarrow$DeiT$\rightarrow$DeiT3). This makes an appropriate exploration of this vast search space essential but computationally very expensive, and we (have to) leave this as an opportunity for future work.
>
> ---
> We hope that our provided answers are able to address all your questions. Please do let us know if there are any remaining concerns, and we will do our best to promptly address these.

---

> > ### Comment · Reviewer_mPnx · 2023-11-22
> >
> > My primary concerns are resolved. Although the BiXT currently does not achieve a compelling performance, it brings new insights for transformers, e.g., disentangling "where" and "what" in the internal representations. It is also understandable that academic researchers have a limited budget. I agree that prioritizing the budget to show the capability for coping with long sequences instead of building larger models is a reasonable choice.
> >
> > Therefore, I would like to raise my rating from 5 (marginally below the acceptance threshold) to 6 (marginally above the acceptance threshold).

---

> > > ### Author Response · Authors · 2023-11-22
> > >
> > > Thank you very much for the response. We appreciate the increase of the score, and the thorough and helpful feedback towards our work.

---

### Author Response · Authors · 2023-11-18
**General response**

Dear reviewers and AC,

We want to genuinely thank you for your valuable time and effort spent reviewing our manuscript, and are grateful for the detailed and constructive remarks that will help us to further improve the quality of our paper.

As the reviewers highlighted, we believe our *BiXT* provides a way to efficiently process longer sequences in an innovative (*qx3u*), neat (*mPnx*) and architecturally simplistic manner (*qx3u, mPnx*), with its effectiveness and versatility validated on four applications across two different modalities (*mPnx*) at a fraction of the compute (*qx3u*) compared to other general Perceiver-like architectures.
We are happy to hear that the reviewers perceived our manuscript as well-written and easy to follow (*mPnx, qx3u, bDbJ*), appreciate the way we motivated the underlying ideas (*qx3u*) and enjoyed reading our paper (*qx3u*).


We individually address each reviewer's comments as direct replies to their respective review, and have uploaded a revised version of the manuscript accommodating the suggested changes (incl. new experimental results and further qualitative insights).

It is of course possible that we might have misinterpreted a comment or question, in which case we would cordially ask the reviewers to point this out to us so we can clarify any remaining points as promptly as possible.

Thank you very much!
The Authors

---

### Meta-Review · Area_Chair_PrhP · 2023-12-09

**Metareview:**

The meta-reviewer has carefully read the paper, reviews, rebuttals, and discussions between authors and reviewers. The meta-reviewer agrees with the reviewers that this submission is a bit below the bar of ICLR. The paper introduces a bi-directional cross-attention Transformer (BiXT), designed to efficiently process long sequences using a small set of latent vectors for the 'what' and input tokens for the 'where'. BiXT's core feature, the bi-directional cross-attention module, refines latent vectors and input tokens while reducing computational and memory demands by exploiting the symmetry of attention patterns. However, the two slightly positive (score 6) reviewers raised many concerns regarding the clarity of the presentations, tables, and backbone architecture. Comparisons to some of the most recent and relevant vision backbones are missing. Additionally, the paper primarily focused on the low-FLOP regime, leaving questions about its applicability to more resource-intensive scenarios. The meta-reviewer feels uncomfortable recommending acceptance, given the current form of the paper. The authors are encouraged to polish and submit the paper to the next venue.

**Justification For Why Not Higher Score:**

N/A

**Justification For Why Not Lower Score:**

N/A

---

### Decision · Program_Chairs · 2024-01-16

Reject